# WisdoM: Improving Multimodal Sentiment Analysis by Fusing Contextual World Knowledge

Wenbin Wang*
School of Computer Science
Wuhan University
Wuhan, China
wangwenbin97@whu.edu.cn

Liang Ding*
The University of Sydney
Sydney, Australia
liangding.liam@gmail.com

Li Shen
Sun Yat-Sen University
Shenzhen, China
mathshenli@gmail.com

Yong Luo†
Wuhan University &
Hubei Luojia Laboratory
Wuhan, China
luoyong@whu.edu.cn

Han Hu
Beijing Institute of Technology
Beijing, China
hhu@bit.edu.cn

Dacheng Tao
Nanyang Technological University
Singapore, Singapore
dacheng.tao@ntu.edu.sg

## Abstract

Multimodal Sentiment Analysis (MSA) focuses on leveraging multimodal signals for understanding human sentiment. Most of the existing works rely on superficial information, neglecting the incorporation of contextual world knowledge (*e.g.,* background information derived from but beyond the given image and text pairs), thereby restricting their ability to achieve better multimodal sentiment analysis (MSA). In this paper, we propose a plug-in framework named WisdoM📖 , to leverage the contextual world knowledge induced from the large vision-language models (LVLMs) for enhanced MSA. WisdoM utilizes LVLMs to comprehensively analyze both images and corresponding texts, simultaneously generating pertinent *context*. Besides, to reduce the noise in the context, we design a training-free contextual fusion mechanism. We evaluate our WisdoM in both the *aspect-level* and *sentence-level* MSA tasks on the Twitter2015, Twitter2017, and MSED datasets. Experiments on three MSA benchmarks upon several advanced LVLMs, show that our approach brings consistent and significant improvements (up to +6.3% F1 score). Code is available at https://github.com/DreamMr/WisdoM.

## CCS Concepts

• **Information systems** → *Sentiment analysis*.

## Keywords

multimodal sentiment analysis; large vision-language model; contextual world knowledge; contextual fusion

---

*Equal contribution.
†Corresponding author.

---

**ACM Reference Format:**
Wenbin Wang, Liang Ding, Li Shen, Yong Luo, Han Hu, and Dacheng Tao. 2024. WisdoM: Improving Multimodal Sentiment Analysis by Fusing Contextual World Knowledge. In *Proceedings of the 32nd ACM International Conference on Multimedia (MM '24), October 28-November 1, 2024, Melbourne, VIC, Australia.* ACM, New York, NY, USA, 10 pages. https://doi.org/10.1145/3664647.3681403

## 1 Introduction

Sentiment analysis (SA) aims to identify the polarity of human sentiment [8, 33, 46]. With the prevalence of social networks, people express their sentiments using not only plain text, but also other modalities of data (*e.g.,* images). Predicting the sentiments using only text data is challenging since the texts are often short and informal on social networks (such as Twitter), while the associated images can provide valuable complementary information. Therefore, it is beneficial to combine multiple modalities for accurate sentiment classification, and multimodal sentiment analysis (MSA) has attracted much attention in recent years [32, 44]. Recent studies improve MSA by carefully designing various strategies, which can be categorized into four types: 1) disentangled representation learning [16, 48], 2) attention-based cross-modal interactions [20, 49, 60], 3) fusion mechanisms [13, 15, 54], and 4) well-designed auxiliary tasks [27, 28, 48].

Despite their empirical success, the above studies only consider the *superficial information*[1] between image and text (see the case "*Aleppo*" in Fig. 1), and sometimes it is difficult to predict the true polarity without their background world knowledge ("*Aleppo has been severely affected by the ongoing Syrian Civil War...*" induced from the large vision-language models). This raises the following question:

### Could world knowledge boost MSA?

Take the test case in Fig. 1 as an example, given a comparative image at different periods alongside a sentence, it is required to answer the question: *What's the sentiment polarity of "Aleppo"?* We employ the current state-of-the-art (SOTA) MSA model [60] as the backbone. As expected, even the SOTA model gives the wrong

---

[1] only reflects the surface or literal information without considering their deep (*e.g.,* historical and cultural) meaning.

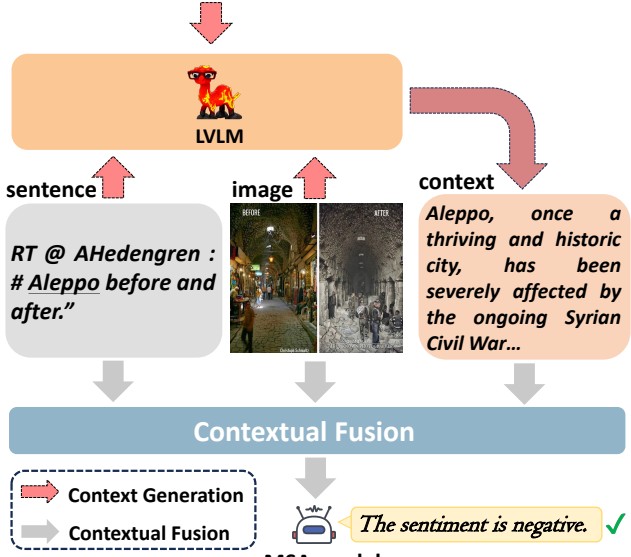

**prompt template**
*Give you a sentence and image, you can provide historical context, important events, and relevant background information related to sentence and image.*

**LVLM**

sentence

*RT @ AHedengren : # Aleppo before and after."*

image

context

*Aleppo, once a thriving and historic city, has been severely affected by the ongoing Syrian Civil War...*

**Contextual Fusion**

→ Context Generation
→ Contextual Fusion

**MSA model**

*The sentiment is negative.* ✓

**Figure 1: The simple schematic of our method. The sentiment polarity of Aleppo is negative, which is difficult to predict directly using existing methods while our WisdoM🟧 predicts correctly via incorporating *context* generated by the world knowledge-rich LVLMs.**

prediction (*i.e.,* "neutral" rather than the groundtruth– "negative") due to the lack of deeper knowledge of "Aleppo" (which is a city in Syria, in conjunction with the image, we might infer that the difference between the before and after of this city is caused by the Syrian war). Therefore, employing world knowledge is essential.

This motivates us to propose a plug-and-play framework termed WisdoM🟧 to utilize the contextual world knowledge (simply induced from the large vision-language models) to complement the existing text-image pair with only superficial information. In particular, our method consists of three main stages: ❶ Prompt Templates Generation, ❷ Context Generation, and ❸ Contextual Fusion. In **stage 1**, we ask language models (*e.g.,* ChatGPT[2]) to generate prompt templates (See the Prompt Template " *Give you a sentence and image, you can ...* " in Fig. 1) which are used to construct instructions for **stage 2**. Then, we employ the advanced large vision-language model (LVLM, *e.g.,* LLaVA [30]) in **stage 2** to generate the contextual information (See the Context " *Aleppo, ... ongoing Syrian Civil War...* " in Fig. 1) based on the provided image and sentence. Note that, we refer to this contextual informations as *context*. Since the derived *context* often contains some noise, we further introduce a novel, training-free Contextual Fusion mechanism in **stage 3**. This mechanism selectively integrates world knowledge by identifying hard samples and fusing them with

---

[2]https://chat.openai.com

*context*, thereby enhancing the model's effectiveness in MSA, particularly on the hard samples that current models may fail.

We validate our WisdoM on several benchmarks, including Twitter2015, Twitter2017 [52] and MSED [18] over several models: LLaVA-v1.5 [29], MMICL [57], Qwen-VL [3], AoM [60], and ALMT [56]. The results across diverse granularities of MSA tasks consistently demonstrate that our approach has substantial improvements (on average +2.2% in terms of F1 score) over the state-of-the-art approaches.

To summarize, our main **contributions** are:

- We propose a plug-in framework WisdoM🟧 , which leverages LVLM to generate explicit contextual world knowledge, to enhance multimodal sentiment analysis.
- To achieve wise knowledge fusion, we introduce a novel contextual fusion mechanism to mitigate the impact of noise in the *context*.
- We conduct extensive analyses and provide some insights on when and why our method works.

## 2 Related Work

### 2.1 Multimodal Sentiment Analysis

Multimodal Sentiment Analysis (MSA), diverging from conventional text-based approaches [17], incorporates diverse modalities (*e.g.,* image) to enhance sentiment classification accuracy [38]. Numerous advanced models have been proposed, covering different levels of granularity, such as sentence and aspect:

*Sentence-Level MSA.* Zhao et al. [58] explore image-text correlations in movie reviews. Li et al. [25] propose a ConvTransformer, blending Transformer [39] and CNN technologies for sentiment analysis. Das and Singh [7] propose a multi-stage multimodal method for the Assamese language, leveraging both text and images. Zhang et al. [56] present an advanced model, ALMT, enhancing multimodal analysis focusing on language-guided features to handle irrelevant or conflicting data across different modalities.

*Aspect-Level MSA.* Yu and Jiang [52] introduce TomBERT, an aspect-oriented multimodal BERT model using annotated tweet datasets. Khan and Fu [21] propose a two-stream model combining an object-aware transformer with non-autoregressive generation [9, 47] for image translation. Zhou et al. [60] present AoM, an aspect-oriented network aimed at reducing distractions in complex image-text interactions.

Although existing approaches relying on sophisticated techniques have achieved remarkable performance in MSA, their limitation lies in relying on superficial information, without incorporating contextual world knowledge.

### 2.2 Large Vision-Language Models

Large Vision-Language Models (LVLMs) are becoming a fundamental tool for solving general tasks [1, 4, 6, 11, 24, 30, 40, 51, 57, 61]. Liu et al. [29] introduce LLaVA, integrating the CLIP ViT-L/14 visual encoder [10] with Vicuna large language model via a projection matrix and two-stage instruction tuning. Zhao et al. [57] propose MMICL to improve LVLMs handling of multi-modal prompts from model and data perspectives. Bai et al. [3] introduce Qwen-VL which is designed to perceive and understand both texts and images.

Here, we leverage LVLMs to enhance multimodal sentiment analysis by generating relevant world knowledge. Additionally, we introduce a new training-free module named Contextual Fusion, designed to minimize noise in the context.

## 2.3 Retrieval-Augmented Generation

Retrieval-Augmented Generation (RAG) improves Large Language Models (LLMs) by adding retrieved text, enhancing performance in knowledge-based tasks [12, 14]. Traditional RAG [23], or Naive RAG, helps in generation but struggles with inconsistent retrieval quality, inaccurate responses, and integrating retrieved context. Advanced methods like DSP [22] enhance context interaction between LLMs and retrieval models, and PKG [31] enables LLMs to access pertinent information for complex tasks without training.

The working mechanism of our WisdoM is similar to RAG, but **different** in the following aspects: ① WisdoM utilizes LVLM to generate world knowledge to provide coherent and accurate context rather than retrieval, ② WisdoM incorporates a contextual fusion mechanism to diminish noise within the context. For additional experimental analysis and discussion, please refer to § 4.4.3.

## 3 Methodology

### 3.1 Preliminary

We first describe the notation of the MSA, then review two typical frameworks for modelling the MSA tasks, where we experiment with our schema upon them: task-specific framework [56, 60] and general-purpose framework [3, 29, 57].

*Notation.* Let $\mathcal{M}$ be a set of multimodal samples. Each sample $m_i \in \mathcal{M}$ consists of a sentence $s_i$ and image $v_i$. For *aspect*-level MSA tasks, there are several aspects $a_i$ which is a subsequence of $s_i$, *i.e.*, $a_i \in s_i$. We denote $f(\cdot)$ as the sentiment classifier. The output of $f(\cdot)$ is the sentiment polarity $y_i \in \{negative, neutral, positive\}$, with corresponding predicted probability denoted as $P_i = \{p_i^{neg}, p_i^{neu}, p_i^{pos}\}$.

*Task-Specific Framework.* For *aspect*-level MSA tasks, the goal is to predict the sentiment polarity $y_i$ and probability $P_i$ for the specific aspect $a_i$ conditioned on the $(v_i, s_i)$, *i.e.*, $(y_i, P_i) = f(v_i, s_i, a_i)$. For *sentence*-level MSA tasks, the $y_i$ and $P_i$ are predicted by sentence $s_i$ alongside the image $v_i$, *i.e.*, $(y_i, P_i) = f(v_i, s_i)$.

*General-Purpose Framework.* To verify that our WisdoM works well on arbitrary architectures, we also apply WisdoM to general-purpose LVLMs. We follow Wang et al. [45] to construct the task instructions $I_{aspect}$ and $I_{sentence}$ for each task to elicit its ability to the corresponding task. The task instructions are presented as single-choice questions with well-formatted options (shown in Table 1). For each choice of a question, we compute the likelihood $P_i$ that LVLM generates the content of this choice based on the given question. The choice with the highest probability is then selected as the prediction $y_i$. The LVLMs can be seen as a sentiment classifier $f(\cdot)$. Thus, the *aspect*-level task and *sentence*-level task can be formulated as $(y_i, P_i) = f(I_{aspect}, v_i, s_i, a_i)$ and $(y_i, P_i) = f(I_{sentence}, v_i, s_i)$ respectively.

### 3.2 WisdoM📙

In this part, we first provide a comprehensive overview of our method, and then introduce it in detail.

**Table 1: Template of task instruction. NOTE: "[sentence]" and "[aspect]" are placeholders meant to be replaced with specific sentence and aspect from the dataset.**

| *Aspect-Level* Taks Instruction $I_{aspect}$ |
| --- |
| Sentence: **[sentence]** Use the image as a visual aids to help you answer the question. What is the sentiment polarity of the aspect **[aspect]** in this sentence? |
| A). positive |
| B). neutral |
| C). negative |
| Answer with the option's letter from the given choices directly. |

| *Sentence-Level* Taks Instruction $I_{sentence}$ |
| --- |
| Sentence: **[sentence]** Use the image as a visual aids to help you answer the question. Given the sentence and image, what is the sentiment conveyed? |
| A). positive |
| B). neutral |
| C). negative |
| Answer with the option's letter from the given choices directly. |

*3.2.1 Overview.* Fig. 2 illustrates the overview of our method following three stages. In the ❶ Prompt Templates Generation, we use large language models, particularly ChatGPT, to provide prompt templates. These prompt templates are fed into the LVLM with sentence $s_i$ and image $v_i$ to generate *context*, also called the ❷ Context Generation. During ❸ Contextual Fusion, we first compute the confidence, determining if the sample is uncertain (referred to as a hard sample). For hard samples, we fuse the predicted probability $P_i$ with $\hat{P}_i$ which is obtained by incorporating *context*. Otherwise, we use $P_i$ as the final prediction.

*3.2.2 Stage 1: Prompt Templates Generation.* The main purpose of this stage is to design the prompt templates used to generate the *context*, so that the LVLM can better understand our intention and thus provide a more comprehensive contextual world knowledge. Inspired by [19, 59], we ask ChatGPT to provide the appropriate prompt templates. The prompt templates provided by ChatGPT consider world knowledge from different perspectives, including historical, social, cultural, *etc*. We insert a "Sentence: [x]" at the end of the prompt template to place the input sentence $s_i$. The example of prompt templates are shown in Supp. A.4.

*3.2.3 Stage 2: Context Generation.* In the context generation stage, prompt templates in ❶ are used to generate *context* that explicitly incorporates world knowledge based on given the image $v_i$ and sentence $s_i$ by LVLMs [30, 51]. Specifically, we construct instruction by replacing the "[x]" in the prompt template with the sentence $s_i$. In addition, different LVLMs require a special token to indicate where the image $v_i$ is inserted. Taking LLaVA [30] as an example, we insert a special token "<image>" at the beginning of the instruction.

*3.2.4 Stage 3: Contextual Fusion.* We use the *sentence*-level task as an example. After obtaining the *context*, we can intuitively use predicted sentiment polarity $\hat{y}_i$ obtained by incorporating *context*,

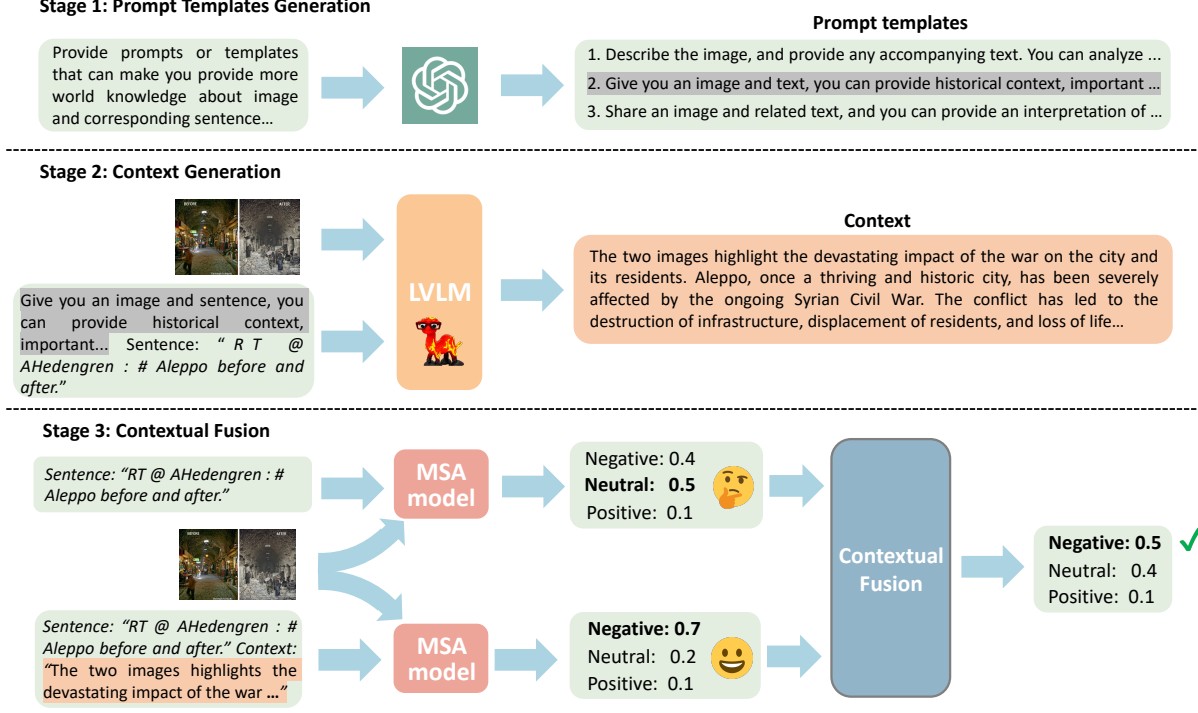

**Figure 2: Detailed illustration of our proposed schema WisdoM🟨 with a running example.** ① **Using ChatGPT to provide prompt templates.** ② **We then prompt LVLMs to generate *context* using the prompt templates with image and sentence.** ③ **A training-free mechanism Contextual Fusion mitigates the noise in the *context*.**

*i.e.,* $(\hat{y}_i, \hat{P}_i) = f(v_i, s_i, context)$. However, the *context* may contain irrelevant information that could disturb performance. Therefore, we first determine hard samples and then fuse the $P_i$ and $\hat{P}_i$ in the hard sample.

***Determining the Hard Samples.*** Inspired by Zhang et al. [55], we find that the ambiguous hard sample is commonly found around boundary areas of the sentiment polarity (*e.g.,* the boundary areas of negative and neutral). Therefore, given a sample $m_i$, we only consider the difference $\delta_i$ between the highest and the second highest probabilities to determine whether it is a hard sample:

$$\delta_i = P_i^{max} - P_i^{sec}, \tag{1}$$

where $P_i^{max}$ and $P_i^{sec}$ represent the highest and second highest probabilities respectively. Then, we denote uncertain threshold $\alpha$ to select samples that do not exceed $\alpha$ as hard, *i.e.,* $\mathcal{V}_{hard} = \{m_i | \delta_i \leq \alpha\}$.

***Fusion with Context.*** Inspired by [26, 35], we take the convex combinations of $P_i$ and $\hat{P}_i$ to obtain the final prediction $\tilde{P}_i$ for hard sample $m_i \in \mathcal{V}_{hard}$:

$$\tilde{P}_i = P_i + \beta \cdot (\hat{P}_i - P_i), \tag{2}$$

where $\beta$ is an interpolation coefficient. Intuitively, $(\hat{P}_i - P_i)$ represents the information incorporated by *context*. $\beta$ is used to control the proportion of information introduced in *context*. When $\beta \rightarrow 0$, the effect brought by *context* is completely ignored and vice versa. Note that, we use $(y_i, P_i)$ as the final prediction when $m_i \notin \mathcal{V}_{hard}$. We study the impact of $\alpha$ and $\beta$ in § 4.2.

## 4 Experiments

In this section, we apply WisdoM to *aspect*-level and *sentence*-level MSA tasks to verify its effectiveness and conduct extensive analysis to better understand the proposed method.

### 4.1 Experimental Settings

*4.1.1 Datasets.* For aspect-level tasks, our two benchmark datasets are Twitter2015 and Twitter2017 [52]. Twitter2015 and Twitter2017 are comprised of multimodal tweets, where each tweet incorporates textual content, an accompanying image, aspects contained within the tweet, and the sentiment associated with each aspect. Each aspect is assigned a label from the predefined set {negative, neutral, positive}. For sentence-level tasks, we evaluate our WisdoM on MSED [18] dataset, containing 9,190 text-image pairs. We show the statistics of Twitter2015, Twitter2017 and MSED in Supp. A.1.

*4.1.2 Models.* To demonstrate WisdoM generalizes across architectures and sizes, we conduct experiments on several models, including MMICL (14B) [57], LLaVA-v1.5 (13B) [29], Qwen-VL (9.6B) [3], ALMT (112.5M) [56], AoM (105M) [60]. The detailed model cards can be found in Supp. A.2.

*4.1.3 Implement Details.* In stage 1, we utilize ChatGPT to generate the prompt templates corresponding to different types of world knowledge, executing this process only once. For Twitter2015 and Twitter2017 datasets, we employ the historical prompt template (see

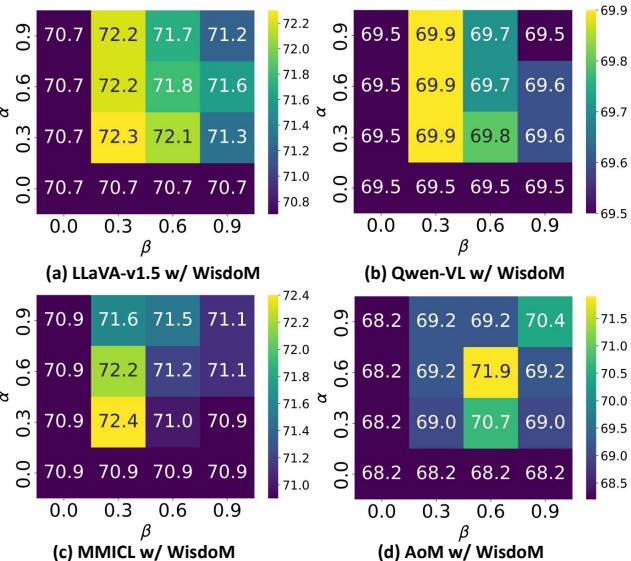

**Figure 3: F1 score on Twitter2015 dev set with different hyper parameters upon several models. The condition $\beta = 0$ implies that the *context* is completely ignored.**

Supp. A.4) to generate *context*. For MSED dataset, the scientific prompt template is utilized for context generation.

*4.1.4 Evaluation Metrics.* For *aspect*-level tasks, we use Accuracy (*Acc.*) and macro-F1 (*Mac-F1*) following the previous studies [20, 21, 28, 60]. For the *sentence*-level task, we adopt precision, recall, and macro-F1 (*Mac-F1*) as evaluation metrics. Following previous work [20], we also perform the paired t-test using scipy[3] to test the significance of the difference between our WisdoM and baseline, with a default significant level of 0.05. To avoid large variances in the final results due to differences in context, we set the temperature to 0.2 for LVLM in stage 2. Additionally, we conduct each experiment five times and report the average of 5 runs for all our experimental results.

## 4.2 Hyper-parameter Selection

In stage 3, Contextual fusion has two major hyper-parameters interpolation coefficient $\beta$ and uncertain threshold $\alpha$. To investigate whether our approach is robust to different hyper-parameters, we employ grid search to study the effect of hyper-parameters on Twitter2015 dev set. As shown in Fig. 3, we find that: 1) although the performance varies with hyper-parameters, the extreme values of the results are not significant. 2) $\alpha$ and $\beta$ values within the range of $[0.3, 0.6]$ demonstrate strong performance among various models. 3) Excessive values of $\beta$ result in performance degradation. We conjecture that this decline may be caused by introducing excessive noise within the *context*. To reduce the redundant adjustment with hyper-parameters, we set $\alpha = 0.3$ and select the optimal value of $\beta$ on the dev set for evaluation.

―――――――――
[3]https://www.scipy.org/

**Table 2: Comparison of our method (upon several advanced models) with existing works on Twitter2015 and 2017 benchmarks. The highest results are highlighted in bold, and * indicates the reproduced results. ± are standard deviations across five runs.**

| Method | Twitter2015 | | Twitter2017 | |
|---|---|---|---|---|
| | *Acc.* | *Mac-F1* | *Acc.* | *Mac-F1* |
| Res-MGAN [21] | 71.7 | 63.9 | 66.4 | 63.0 |
| Res-BERT+BL [21] | 75.0 | 69.2 | 69.2 | 66.5 |
| mPBERT(CLS) [21] | 75.8 | 71.1 | 68.8 | 67.1 |
| ESAFN [53] | 73.4 | 67.4 | 67.8 | 64.2 |
| TomBERT [52] | 77.2 | 71.8 | 70.3 | 68.0 |
| CapTrBERT [21] | 77.9 | 73.9 | 72.3 | 70.2 |
| JML [20] | 78.7 | - | 72.7 | - |
| VLP-MABSA [28] | 78.6 | 73.8 | 73.8 | 71.8 |
| CMMT [49] | 77.9 | - | 73.8 | - |
| MMICL [57]* | 76.0 | 72.7 | 74.1 | 74.0 |
| -w/ WisdoM🦉 | 77.3±0.1 | 74.2±0.2 | 75.7±0.1 | 75.7±0.1 |
| LLaVA-v1.5 [29]* | 77.9 | 74.3 | 74.6 | 74.3 |
| -w/ WisdoM🦉 | 78.9±0.3 | 75.6±0.3 | 75.6±0.4 | 75.3±0.3 |
| Qwen-VL [3]* | 75.0 | 70.0 | 71.9 | 71.2 |
| -w/ WisdoM🦉 | 76.2±0.1 | 71.3±0.2 | 73.8±0.2 | 72.8±0.3 |
| AoM [60]* | 80.0 | 75.2 | 75.9 | 74.5 |
| -w/ WisdoM🦉 | **81.5**±0.1 | **78.1**±0.2 | **77.6**±0.4 | **76.8**±0.2 |

## 4.3 Main Results

*4.3.1 Results of Aspect-Level MSA Task.* We compare against advanced aspect-level MSA methods on Twitter2015 and Twitter2017, and report the results on Table 2. We show that our WisdoM achieves consistent and significant improvement on four models across two datasets. The WisdoM brings max 2.9% and 2.3% F1-gains on Twitter2015 and Twitter2017 respectively, showing that our method has a clear advantage. Besides, we perform the paired t-test to assess the significance of the improvements made by our WisdoM over the baseline. The results demonstrate that our improvement is statistically significant with $p - value < 0.05$.

*4.3.2 Results of Sentence-Level MSA Task.* As shown in Table 3, notably, our WisdoM upon Qwen-VL achieves the **new SOTA** F1 score: 91.8%, outperforming Qwen-VL (89.4%), LLaVA-v1.5 (88.8%), MMICL (86.2%) and ALMT (83.7%), consistently. The most significant improvement is achieved on ALMT (112.5M), where we bring an encouragingly 6.3% F1 gain, suggesting that contextual world knowledge is particularly crucial for small models in MSA. Besides, our WisdoM is statistically significant with $p - value < 0.05$ when comparing with baseline.

## 4.4 Ablation Study

*4.4.1 Impact of different modules.* To better understand the role of each module in our WisdoM, Table 4 presents the ablation results of the gradual addition of different components. Compared with the baselines (MMICL and LLaVA-v1.5), only adding *context* results in a slight performance degradation (-0.3% average F1 score), while with the help of our proposed Context Fusion mechanism, we achieve a consistent and significant improvement (+1.8% average F1 score). Through (error) case studies in Supp. C, we find that

**Table 3: Performance of applying our WisdoM to advanced models on MSED benchmark, with reference results from existing works. The best results are bolded, and the * denotes the reproduced results. ± are standard deviations across five runs.**

| Method | MSED | | |
|---|---|---|---|
| | *Precision* | *Recall* | *Mac-F1* |
| DCNN [18] | 59.3 | 53.0 | 51.2 |
| BiLSTM [18] | 78.4 | 78.8 | 78.6 |
| DCNN+AlexNet [18] | 71.0 | 70.1 | 70.3 |
| DCNN+ResNet [18] | 74.7 | 74.7 | 74.6 |
| BiLSTM+AlexNet [18] | 78.7 | 79.2 | 78.9 |
| BiLSTM+ResNet [18] | 75.9 | 75.3 | 75.3 |
| BERT+AlexNet [18] | 83.2 | 83.1 | 83.2 |
| Multimodal Transformer [18] | 83.6 | 83.5 | 83.5 |
| ALMT [56]* | 83.7 | 84.0 | 83.7 |
|   -w/ WisdoM📙 | 89.9±0.7 | 90.1±0.9 | 90.0±0.8 |
| MMICL [57]* | 86.2 | 86.6 | 86.2 |
|   -w/ WisdoM📙 | 89.9±0.3 | 88.2±0.4 | 88.9±0.3 |
| LLaVA-v1.5 [29]* | 89.0 | 88.8 | 88.8 |
|   -w/ WisdoM📙 | 90.6±0.1 | 90.4±0.3 | 90.5±0.2 |
| Qwen-VL [3]* | 90.5 | 88.9 | 89.4 |
|   -w/ WisdoM📙 | **91.8**±0.2 | **91.7**±0.2 | **91.8**±0.2 |

**Table 4: Ablation study of *context* and its wise fusion module. "CF" denotes our Contextual Fusion. We first only incorporate "*context*" and subsequently introduce the "contextual fusion" module.**

| Method | Twitter2017 | | MSED | | *Avg.* |
|---|---|---|---|---|---|
| | *Acc.* | *Mac-F1* | *Recall* | *Mac-F1* | |
| | | MMICL | | | |
| Baseline | 74.1 | 74.1 | 85.9 | 86.2 | 80.1 |
| + *context* | 72.6 (-1.5) | 74.4 (+0.3) | 87.6 (+1.7) | 86.8 (+0.6) | 80.4 |
| + CF | **75.7** (+1.6) | **75.7** (+1.6) | **89.1** (+3.2) | **88.9** (+2.7) | **82.4** |
| | | LLaVA-v1.5 | | | |
| Baseline | 74.6 | 74.3 | 89.1 | 88.8 | 81.7 |
| + *context* | 73.7 (-0.9) | 73.5 (-0.8) | 87.8 (-1.3) | 87.7 (-1.1) | 80.7 |
| + CF | **75.6** (+1.0) | **75.3** (+1.0) | **90.6** (+1.5) | **90.5** (+1.7) | **83.0** |

containing irrelevant information in the original context leads to bad performance, showing the necessity of further context fusion mechanism.

*4.4.2 How does hand-crafted prompt template differ from LLM-generated in stage 1?* To compare human-crafted prompt template and LLM-generated prompt template, we employ three human annotators and ChatGPT to provide prompt templates respectively. We provide the same meta-prompt (*"Please provide prompt template that large vision-language model can generate the historical knowledge based on the image and sentence."*) for human annotators and ChatGPT to generate prompt templates. The details of prompt templates can be seen in Supp. A.6. As shown in Table 5, the results show that LLM-generated consistently outperformed

**Table 5: Comparison between hand-crafted templates and LLM-generated templates on Twitter2015 and Twitter2017. We report the results of hand-crafted and LLM-generated averaged across 3 separate prompts.**

| Model | Type | Twitter2015 | | Twitter2017 | |
|---|---|---|---|---|---|
| | | *Acc.* | *Mac-F1* | *Acc.* | *Mac-F1* |
| MMICL | Hand-crafted | 77.2 | 74.0 | 75.4 | 75.3 |
| | **LLM-generated** | **77.3** | **74.6** | **75.8** | **75.7** |
| LLaVA-v1.5 | Hand-crafted | 78.3 | 75.4 | 74.8 | 74.9 |
| | **LLM-generated** | **79.6** | **76.2** | **75.3** | **75.2** |
| Qwen-VL | Hand-crafted | 77.4 | 73.1 | 73.0 | 72.1 |
| | **LLM-generated** | 77.4 | **73.2** | **73.3** | **72.5** |
| AoM | Hand-crafted | 81.2 | 77.9 | 77.0 | 76.3 |
| | **LLM-generated** | **81.9** | **78.6** | **77.2** | **76.6** |

**Table 6: Comparative results of *context* generated by Naive RAG, PKG, and our stage 2. We incorporate the *contexts* on LLaVA-v1.5 directly.**

| Method | Twitter2015 | | Twitter2017 | |
|---|---|---|---|---|
| | *Acc.* | *Mac-F1* | *Acc.* | *Mac-F1* |
| Naive RAG | 75.1 | 71.0 | 71.9 | 70.7 |
| PKG | 76.2 | 72.4 | 72.8 | 71.5 |
| **Our Context** | **76.3** | **72.7** | **73.7** | **73.5** |

hand-crafted across various metrics. Furthermore, the standard deviation for hand-crafted templates is 1.1, compared to only 0.3 for LLM-generated templates. We find that ***templates provided by human annotators directly seek historical insights from image and text, while templates generated by ChatGPT offer deeper analysis, reflecting important events, significance and background information.***

*4.4.3 How does context generation in stage 2 compare to that retrieved by RAG-based methods?* To further analyse the effect of *context*, we compare our *context* generated in **stage 2** with the document retrieved by Naive RAG [23] and knowledge generated by PKG [31], collectively termed as "context" for simplicity. The assessment focuses on the context's pertinence to a given image $v$ and sentence $s$, alongside its applicability in MSA tasks. We employ the LLM-based metric, *i.e.,* **LLM-as-a-Judge** [5] to quantify the quality of *context*. Specifically, we craft a prompt for GPT-4V [36] to compare our context with that provided by Naive RAG and PKG. The detailed experimental settings can be found in Supp. A.7. As shown in Fig. 4, our context significantly beats the Naive RAG and PKG counterparts, demonstrating its superiority. We provide examples of context provided by different methods in Supp. C.1. Besides analyzing the contexts' pertinence of different methods, we report their downstream performance on MSA tasks in Table 6. Clearly, the MSA performance with our context is the best. The results above illustrate that ***our method can provide more precise context, thus bringing better MSA performance.***

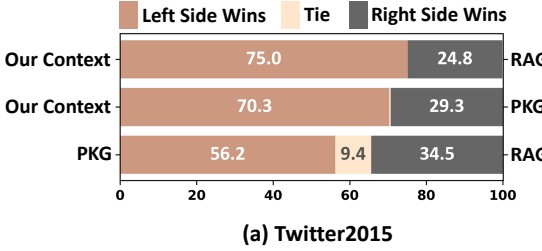

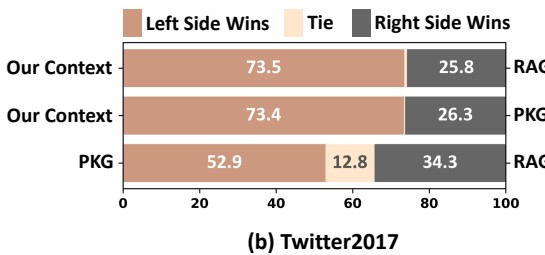

**Figure 4: Comparative winning rates of Our Context *v.s.* RAG-based methods on Twitter2015 and Twitter2017 benchmarks. We can see that our contexts are better than the knowledge provided by (Naive) RAG and PKG.**

**Table 7: Ablation study of different fusion strategies. "JS" denotes Jensen-Shannon divergence. "CF" denotes our Contextual Fusion.**

| Method | Twitter2015 | | Avg. |
|---|---|---|---|
| | *Acc.* | *Mac-F1* | |
| AoM | 80.0 | 75.2 | 77.6 |
| -w/ Average | 80.3 (+0.3) | 70.7 (-4.5) | 75.5 (-2.1) |
| -w/ Max | 80.6 (+0.6) | 76.8 (+1.6) | 78.7 (+1.1) |
| -w/ JS | 80.8 (+0.8) | 77.4 (+2.2) | 79.1 (+1.5) |
| -w/ CXMI | 79.7 (-0.3) | 75.7 (+0.5) | 77.7 (+0.1) |
| **-w/ CF** | **81.5** (+1.5) | **78.1** (+2.9) | **79.8** (+2.2) |
| MMICL | 76.0 | 72.7 | 74.4 |
| -w/ Average | 75.5 (-0.5) | 72.3 (-0.4) | 73.9 (-0.5) |
| -w/ Max | 75.4 (-0.6) | 72.2 (-0.5) | 73.8 (-0.6) |
| -w/ JS | 77.1 (+1.1) | 73.9 (+1.2) | 75.5 (+1.1) |
| -w/ CXMI | 76.3 (+0.3) | 73.4 (+0.7) | 74.8 (+0.4) |
| **-w/ CF** | **77.3** (+1.3) | **74.2** (+1.5) | **75.8** (+1.4) |

*4.4.4 How does the Contextual Fusion module compare to other fusion strategies?* In Table 7, we explore different fusion strategies, including $mean(P_i, \hat{P}_i)$ ("**Average**"), $max(P_i, \hat{P}_i)$ ("**Max**"), Jensen-Shannon divergence [34] ("**JS**"), conditional cross-mutual information $f_{cxmi}$ [43] ("**CXMI**"), and our Context Fusion ("**CF**"). For JS, we calculate the JS divergence of $P_i$ with the uniform distribution to serve as the fusion weight, *i.e., $\beta$*. As for CXMI, if $f_{cxmi} > 1.1^4$, we adopt $(y_i, P_i)$ as our ultimate prediction, otherwise, we use $(\hat{y}_i, \hat{P}_i)$ as the final prediction. The results show that *our Contextual Fusion module performs the best among all competitive alternatives, confirming its effectiveness.*

## 4.5 Scalability of WisdoM

Our plug-in method is data- and model-agnostic, and therefore, it is expected to be highly scalable. Here we scale our WisdoM up to different model sizes and data volumes.

*4.5.1 Performance on Different Model Sizes.* We experiment with scaling the model size to see if there are ramifications when operating at a larger scale. Fig. 5 (a) reveals that the performance increases as the LVLM size increases. In addition, we find that *as*

---
[4] In preliminary study, we grid-searched values ranging from 0.5 to 2.0, and 1.1 performs best on the dev set, thus leaving as our default setting.

*the size of the model increased, the performance gains became more pronounced.*

*4.5.2 Performance on different Data Volumes.* We conduct experiments on different ratios of training data to verify the robustness of WisdoM. As shown in Fig. 5 (b), we find that *our WisdoM can consistently bring improvement, even in scenarios with extremely limited training data.*

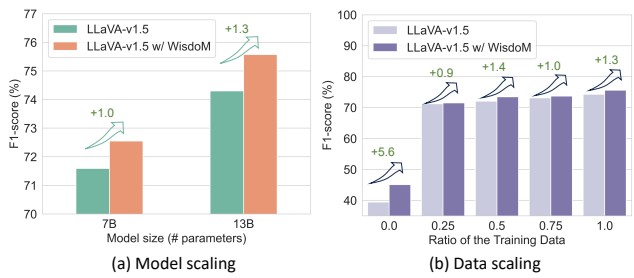

**Figure 5: Performance of scaling WisdoM on Twitter2015 with different a) model sizes and b) data scales.**

## 4.6 Exploring Contexts Derived from Various LVLMs

To explore the relationship between *context* and LVLMs capability, we conduct experiments on AoM using *contexts* derived from mPLUG-Owl2 [51] (8.2B) and LLaVA-v1.5 (13B). As depicted in Table 8, the results show that *the stronger the capability of LVLMs, the more accurate and helpful the generated context is for MSA.*

**Table 8: Comparison of contexts derived from different LVLMs. "Context$_m$" and "Context$_L$" represent the context derived from mPlUG-Owl2 and LLaVA-v1.5 respectively.**

| Method | Twitter2015 | | Twitter2017 | |
|---|---|---|---|---|
| | *Acc.* | *Mac-F1* | *Acc.* | *Mac-F1* |
| mPLUG-Owl2 | 76.8 | 72.3 | 74.2 | 73.0 |
| **LLaVA-v1.5** | **77.9** | **74.3** | **74.6** | **74.3** |
| AoM | 80.0 | 75.2 | 75.9 | 74.5 |
| -w/ Context$_m$ | 81.2 | 77.8 | 76.4 | 75.2 |
| **-w/ Context$_L$** | **81.5** | **78.1** | **77.6** | **76.8** |

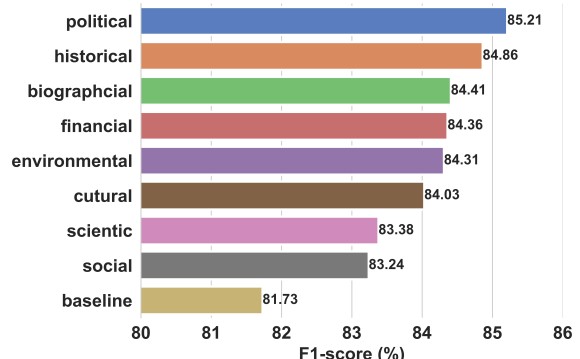

Figure 6: Effects of different types of world knowledge on Politician Twitter Dataset. We analyse the effect of different types of world knowledge by applying WisdoM to AoM. NOTE: "baseline" represents AoM which retrains on Politician Twitter training set.

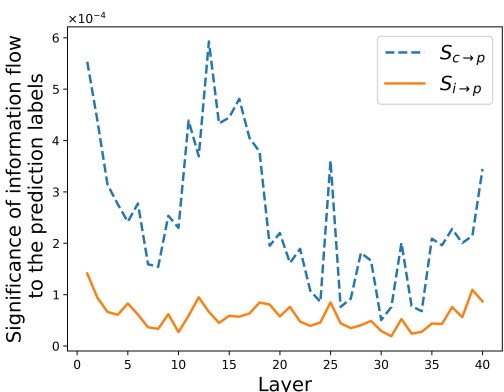

Figure 7: Comparison of context ($S_{c \to p}$) and input's ($S_{i \to p}$) correlation to the final prediction across layers in LLaVA-v1.5 on Twitter2015. High score means a strong correlation with final decision-making.

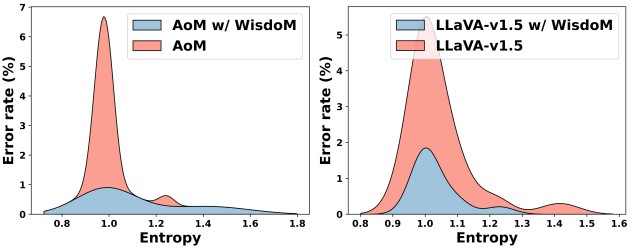

Figure 8: Visualizing of error rate for hard samples ($\delta \leq 0.3$) on Twitter2015 benchmark.

## 4.7 When and Why Does Our Method Work?

To better understand when and why our method works, we conduct extensive analysis to provide the following insights:

***World Knowledge Enhances MSA, while Domain-related Knowledge is more Helpful.*** For better understanding the effects of different types of world knowledge, we conduct experiment with AoM on the Politician Twitter dataset [50], covering 8 perspectives of world knowledge (the prompt templates can be found in Supp. A.4). The Politician Twitter dataset is derived from the Twitter2015 and Twitter2017 datasets, segregating all multimodal tweets that mention politicians like Barack Obama, Donald Trump, and Hillary Clinton into the test set, and randomly distributing the remaining tweets into training and dev set. The detailed statistics of the Politician dataset is shown in Supp. A.1. It is notable that we retrain the AoM on the Politician Twitter dataset to avoid data leakage. Fig. 6 shows that 1) nearly all types of extra knowledge enhance the MSA performance, 2) political knowledge significantly enhances MSA on Politician Twitter dataset (brings 3.48% improvements on F1 score).

***Context is Dominant for Prediction.*** To draw a clearer picture of the information flow for MSA, we calculate $S_{c \to p}$ and $S_{i \to p}$ to represent the mean significance of information flow [37, 41] from *context* (*c*) and original input (*i*) to the prediction labels (*p*) respectively (detailed in Supp. A.8). Fig. 7 reveals that the significance of the information flow from *context* to the prediction label is remarkably higher than its *input* counterpart, suggesting that *context* outweighs image and sentence when making the final prediction.

***WisdoM Effectively Reduces the Uncertainty of Hard Samples.*** To further explore how our WisdoM affects the hard samples, we visualize the error rate within high entropy in Fig. 8. After integrating WisdoM, the error rate is significantly decreased compared with the baseline (AoM and LLaVA-v1.5), demonstrating that our WisdoM effectively reduces the uncertainty of hard samples and improves performance. We hypothesize that part of such eliminated information may be multimodal hallucination [42], while another part may be due to the model's inherent knowledge gaps [2]. Both aspects will be investigated in our future work.

## 5 Conclusion

In this paper, we propose a simple yet effective plug-in framework WisdoM 📙 to enhance the ability of multimodal sentiment analysis. Our WisdoM contains three stages: Prompt Templates Generation, Context Generation, and Contextual Fusion. We empirically demonstrated the effectiveness and universality of the WisdoM on several widely-used benchmarks. From the results, we mainly conclude that: (1) World knowledge can improve MSA, and domain-related knowledge can be very beneficial; (2) A more precise context is more helpful for MSA; (3) World knowledge is particularly desirable for hard samples. In the future, we will develop adaptive mechanisms to dynamically integrate world knowledge into MSA models.

## LIMITATIONS

Our work has potential limitations. We use LVLMs to generate context, but this can lead to hallucinations and incorrect MSA model results. Additionally, our WisdoM relies on LVLMs' internal world knowledge, which requires periodic updates over time. In future work, we aim to investigate adaptive mechanisms for dynamic integration of world knowledge into MSA models.

# Acknowledgments

This work was supported in part by the National Key Research and Development Program of China under No. 2021YFC3300200, the National Natural Science Foundation of China (Grant No. U23A20318), the Fundamental Research Funds for the Central Universities (No. 2042024kf0039), and the Special Fund of Hubei Luojia Laboratory under Grant 220100014. The numerical calculations in this paper were done using the supercomputing system at the Supercomputing Center of Wuhan University.

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
