# OpenReview forum: "WisdoM: Improving Multimodal Sentiment Analysis by Fusing Contextual World Knowledge"
_acmmm.org/ACMMM/2024/Conference — MM2024 Poster_

### Official Review · Reviewer_pkHT · 2024-05-19

**Rating:** 5
**Confidence:** 3

**Summary:**

This paper focuses on multimodal sentiment, and proposes a large vision-language model (LVLM) WisdoM for multimodal aspect-level and sentence-level sentiment analysis. Wisdom leverages the generation capability of ChatGPT to generate prompts and the inference capability of LVLM to retrieval related world knowledge, and introduces a contextual fusion mechanism for achieving final better sentiment analysis results. This paper proves the superiority of WisdoM comparing with related work by experiments.

**Strengths:**

1. The WisdoM model proposed by this paper leverages the generation capability of ChatGPT and the inference capability of different LVLMs. In terms of the application of large (vision) language models, this is a deliberate exploration.
2. In this paper, sufficient experimental design, results presentation and analysis are given.
3. The structure and order of this paper are clear, and the content is readable.

**Limitations:**

1. The issue of sentiment analysis (both sentence-level and aspect-level) itself and the idea of introducing external knowledge are not entirely new, and of course I do not deny the significance of the work in this paper.
2. In stage 3, it has not been given what models for MSA, although this does not affect the understanding of the work in this paper. In addition, how do plug and play reflect?
3. Would it have been better to use a large (vision) language model with prompt to complete the entire sentiment analysis task? If this issue is discussed in this paper, it will be more perfect.

**Suitability:**

3

---

### Official Review · Reviewer_oyfg · 2024-05-25

**Rating:** 4
**Confidence:** 3

**Summary:**

This paper proposes a plug-in framework WisdoM, which leverages LVLM to generate explicit contextual world knowledge, to
enhance multimodal sentiment analysis. To achieve wise knowledge fusion, this paper introduces a contextual fusion mechanism to mitigate the impact of noise in the context. They conduct extensive analyses and provide some insights on when and why their method works

**Strengths:**

The paper's writing is characterized by a clear and coherent style that effectively conveys complex ideas to readers. The proposed methodology is intricate yet well-conceived and logically sound. Moreover, the experimental section is comprehensive, allowing for a detailed evaluation of each model component's functionality and demonstrating superior performance compared to previous baselines.

**Limitations:**

Because of the slow inference speed of LVLMs, the efficiency of the algorithm is a concern. It is recommended to add a time vs. accuracy comparison chart to highlight the efficiency of this method.

**Suitability:**

3

---

### Official Review · Reviewer_XHBh · 2024-05-27

**Rating:** 2
**Confidence:** 3

**Summary:**

The manuscript introduces an innovative framework, WisdoM, which integrates contextual world knowledge derived from large vision-language models (LVLMs) into multimodal sentiment analysis (MSA). This approach is commendable as it attempts to address the limitation of relying solely on superficial data by incorporating deeper, contextual insights, potentially resulting in more accurate sentiment analysis.

**Strengths:**

1. The methodology employed in the study, involving stages like Prompt Templates Generation, Context Generation, and Contextual Fusion, is well-structured and detailed. The use of a plug-and-play framework that can be integrated with existing MSA systems is particularly noteworthy and could have significant implications for practical applications.
2. The evaluation of the WisdoM framework across multiple datasets (Twitter2015, Twitter2017, and MSED) and its comparison with several state-of-the-art models demonstrate a rigorous testing of the proposed solution. The reported improvements in F1 scores add credibility to the claims of enhanced performance.

**Limitations:**

1. The motivation behind the study, as pointed out, appears somewhat unclear or inadequately justified. While the authors argue that existing methods predominantly rely on superficial information, there is a lot of work in sentiment analysis that already incorporates deeper understanding through various sophisticated models. A more thorough literature review and a clearer articulation of how WisdoM distinctly improves over these existing deep analysis methods would strengthen the manuscript.
2. It is crucial for the authors to dissect the sources of the observed performance improvements. The current manuscript does not clearly delineate whether the improvements are primarily due to the use of large models (like LVLMs) or from the novel components introduced in WisdoM.
3. As mentioned above, although the paper mentions the use of advanced LVLMs, it does not provide a detailed comparison with other deep learning techniques that are specifically designed for deep semantic analysis in multimodal contexts. Including such comparisons could highlight the unique benefits or shortcomings of the proposed framework relative to existing deep learning solutions.

**Suitability:**

3

---

### Official Review · Reviewer_TStS · 2024-05-29

**Rating:** 4
**Confidence:** 2

**Summary:**

The paper presents WisdoM, a novel framework that integrates contextual world knowledge into Multimodal Sentiment Analysis (MSA) using Large Vision-Language Models (LVLMs). It features a three-stage process comprising prompt template generation, context generation, and a training-free contextual fusion mechanism to selectively enhance models with robust world knowledge and reduce noise. Experiments on Twitter2015, Twitter2017, and MSED datasets demonstrate significant improvements in sentiment analysis performance, with the potential to dynamically incorporate world knowledge into MSA models.

**Strengths:**

1. **Framework Innovation**: The introduction of WisdoM, a novel plug-in framework that harnesses the generative capabilities of Large Vision-Language Models (LVLMs) to produce explicit contextual world knowledge, significantly enhancing the depth of Multimodal Sentiment Analysis (MSA).

2. **Knowledge Fusion Mechanism**: The development of a training-free Contextual Fusion mechanism that intelligently integrates world knowledge, mitigating the impact of noise in the context and improving the model's effectiveness, particularly for challenging samples that current models may find difficult to analyze.

3. **Empirical Validation and Scalability**: Extensive experimentation and analysis across multiple benchmarks, demonstrating not only the consistent and significant improvements in MSA performance but also the scalability of the WisdoM framework across various model sizes and data volumes, showcasing its potential for dynamic incorporation of world knowledge into sentiment analysis models.

**Limitations:**

1.The most significant issue lies in the potential for hallucinations in the output of LLMs. How can you determine whether the Context produced by LLMs is always beneficial for multimodal sentiment analysis tasks?

2.Over time, the contextual world knowledge may require periodic updates.

3.The paper should include the limitations section.

**Suitability:**

2

---

### Meta-Review · Area_Chair_g55L · 2024-06-29

**Recommendation:** Accept (Poster)
**Confidence:** 5

**Metareview:**

The manuscript introduces WisdoM, an innovative framework that effectively integrates Large Vision-Language Models (LVLMs) to advance Multimodal Sentiment Analysis (MSA) by embedding explicit contextual world knowledge and employing a training-free Contextual Fusion mechanism. This approach enhances the depth of sentiment analysis and demonstrates robust empirical validation and scalability across various benchmarks, underscoring its potential for real-world applications. However, to strengthen the paper further, it is crucial to address concerns such as the clarity on model usage in stage 3, the need for a discussion on LVLMs and prompt-based tasks, and an evaluation of efficiency vis-à-vis LVLMs' slower inference speed. Additionally, clarifying the motivation behind the study and distinguishing between improvements attributed to large models versus novel components would bolster the manuscript's impact. Lastly, considering the potential for LLM output hallucinations and the necessity for periodic updates of contextual world knowledge is critical for ensuring the framework's robustness. These aspects should be carefully addressed in the camera-ready version to enhance the paper's contribution and clarity.